# Heat Illness Requiring Emergency Care for People Experiencing Homelessness: A Case Study Series

**DOI:** 10.3390/ijerph192416565

**Published:** 2022-12-09

**Authors:** Timothy English, Matthew Larkin, Alejandro Vasquez Hernandez, Jennie Hutton, Jane Currie

**Affiliations:** 1Heat and Health Research Incubator, Faculty of Medicine and Health, The University of Sydney, Sydney, NSW 2050, Australia; 2Homeless Health Service, St Vincent’s Hospital, Sydney, NSW 2010, Australia; 3Emergency Department, St Vincent’s Hospital, Melbourne, VIC 3065, Australia; 4School of Nursing, Queensland University of Technology, Brisbane, QLD 4000, Australia

**Keywords:** emergency medical services, emergency service, extreme heat, heatwave, heatstroke, heat illness

## Abstract

Extreme heat and hot weather has a negative impact on human health and society. Global warming has resulted in an increase in the frequency and duration of heatwaves. Heat-related illnesses are a significant negative consequence of high temperatures and can be life-threatening medical emergencies. The severity of the symptoms can depend on the pre-existing medical conditions and vary from mild headaches to severe cases that can lead to coma and death. The risk of heat-related illness may be higher for people experiencing homelessness due to a lack of access to cool places and water, and the complex interactions between mental illness, medications and substance use disorder. This paper presents two cases of people experiencing homelessness who were admitted to the emergency department of a hospital in Sydney, Australia during a heatwave in November 2020. Both cases were adult males with known risk factors for heat-related illness including hypertension and schizophrenia (Case One) and hepatitis C, cirrhosis, and alcohol use disorder (Case Two). These cases show that severe weather can not only be detrimental to homeless people’s health but can also cause a significant economic toll, evident by the $70,184 AUD expenditure on the care for these two cases. This case report highlights the requirement to determine the risk of heat-related illness to people experiencing homelessness and need to protect this vulnerable population from weather-related illness and death.

## 1. Introduction

Global warming has increased the frequency and duration of heatwaves [1,2]. There is no international standard definition of a heatwave but generally, they occur when the maximum and minimum temperatures in a location are unusually high over two or more days, which could be accompanied by elevated humidity [3,4]. Climate change in conjunction with urbanization are reported to be the main extrinsic factors for heatwaves [3,5]. Australia’s number one natural hazard killer is heatwaves, resulting in more deaths than any other natural disaster [6]. The year 2020 was Australia’s fourth warmest year on record at 1.15 °C above the average annual national mean temperature [7]. Heatwaves have become more intense with a consistent increase in peak temperature creating substantial public health challenges [8,9,10]. During heatwaves in Australia there is an increase in emergency department (ED) presentations, hospital admissions, emergency call outs and mortality [11]. Over a 12-year period in Brisbane, Australia, there were 178 heat-related illness presentations to ED during a total of 35 heatwave days [12]. In Perth, Australia, the total hospital costs related to heat over a 6-year period was estimated to be 79.5 million AUD [13].

Extended periods of high ambient temperature place people at risk of heat-related illness. Heatstroke is the most dangerous condition in a spectrum of heat-related illnesses and is characterized by an elevation of core temperature above 40 °C (hyperthermia), accompanied by signs of neurological damage and multisystem failure [14,15]. Heatstroke can be classified as either classic (passive) or exertional. Both types are medical emergencies caused by a failure to compensate for excessive body heat, but the underlying mechanisms are different. Classic heatstroke is caused by exposure to a hot environment for a prolonged time and the body is unable to compensate, whereas exertional heatstroke is caused by physical exertion and occurs when excessive metabolic heat production overwhelms physiological heat-loss mechanisms [14,16].

Adults with coexisting conditions, particularly cardiovascular disease, are at an increased risk of heat-related mortality and morbidity due to an increased cardiovascular strain during heat stress [17,18,19]. Often, heat-related illness incidents are under-reported since they are frequently attributed to cardiovascular disease. Additionally, some medications commonly used for chronic conditions may predispose patients to heat-related complications [20,21]. Medications such as anticholinergics and psychotropic drugs may sensitize a patient to heat by disrupting thermoregulatory responses that maintain core temperature [22,23], however these claims require further experimental research.

The effects of extreme heat may be amplified for those who are homeless. These people are unable to avoid direct exposure to the climate, which may increase the risk of heat-related illness [24]. The 2016 census estimated there were 116,427 people experiencing homelessness in Australia [25], and the economic impact of the COVID-19 pandemic may see this number increase. Most people experiencing homelessness live in metropolitan cities where finding cool space is particularly difficult as the built environment retains heat [24,26]. Lack of access to cool places and water, and the impact of mental illness, medications, and substance use disorder may increase the risk of heat-related illness for this cohort [27].

In recognition of the diversity of homelessness, the definition includes three categories: (1) people who experience *Primary Homelessness* do not have conventional housing and are sleeping rough or in improvised dwellings; (2) people who experience *Secondary Homelessness* frequently move from one temporary shelter to another such as emergency accommodation, refuges or couch surfing; and (3) people who experience *Tertiary Homelessness* live in accommodations that do not meet minimum community standards such as boarding housing and caravan parks [28].

People experiencing homelessness are poorly represented in disaster health planning and there are very few studies available to inform the development of homeless health weather-related illness prevention strategies. The aim of this paper was to report two cases of heat illness requiring emergency care in people experiencing homelessness to highlight the requirement for further research aimed at protecting this vulnerable population from weather-related illness and death.

## 2. Materials and Methods

The study design is a retrospective case study series investigating known heat illness cases that presented to the hospital’s ED over the November 2020 heatwave in Sydney, Australia. Case One presented on 28 November 2020 and Case Two on 29 November 2020. The maximum temperature and relative humidity reported in Sydney at the Observatory Hill weather station, by the Bureau of Meteorology (Australian Government) [29] was 40.8 °C/51% relative humidity on the 28th when Case One was admitted, and 40.5 °C/22% relative humidity on the 29th when Case Two was admitted. This was only the second instance of consecutive days with a maximum ambient temperature ≥40 °C at this location (the only other time was 26 and 27 January 1960) [29]. Following ethics review and approval through the hospital’s Human Research Ethics Committee (2021/ETH01128), the first step of this case series was to identify people experiencing homelessness that presented to ED with heat illness over the designated period. This was achieved by reviewing the admissions of people experiencing homelessness via the Emergency Department Information System (EDIS). Once the cases were located the medical records were extracted and stored as non-identifiable data for analysis. A waiver of consent was awarded by the hospital’s Human Research Ethics Committee to access the data without requiring informed consent from each of the case study participants because they were not contactable at the time of ethical review.

*Data Collection:* The study team collected case information including the location each patient was found when assisted into ED, time of admission to ED, gender, age, pre-existing conditions, the approximate duration of time experiencing homelessness, medications at the time of incident, signs and symptoms, core temperature, blood markers, length of stay in hospital and treatment administered. The team also determined the economic cost to the hospital for each case. Information was accessed from hospital medical records.

## 3. Case Series

### 3.1. Case One

A 66-year-old male with a history of schizophrenia and natural killer (NK) leukaemia was admitted to the ED at the hospital in Sydney, Australia, on 28 November 2020 (time of admission to ED 19:41) after collapsing (no seizure activity noted) in the street while waiting to be served at a mobile food van. Bystanders called an ambulance, and when paramedics arrived, the patient was unresponsive, febrile and breathing spontaneously requiring support via bag-valve-mask (BVM). Case One’s first vital signs in ED were as follows: temperature (T) of 41.3 °C measured with an infrared tympanic thermometer, blood pressure (BP) of 80/–mmHg, heart rate (HR) of 138 bpm, respiratory rate (RR) of 8 and oxygen saturation (SpO_2_) of 99% via 15L BVM. Electrocardiogram (ECG) showed a sinus tachycardia with no ST elevation. Case One was unresponsive to all stimuli with a Glasgow Coma Scale (GCS) score of 3. On examination, Case One had pinpoint pupils, urinary incontinence, no evidence of trauma, and no signs of heart failure on chest X-Ray. Case One was diagnosed with heatstroke. Case One’s main clinical data are presented in Table 1.

The initial management of Case One consisted of intravenous (IV) administration of 800 mcg naloxone with no effect noted, two IV access cannulae, one litre (L) of cold IV fluid resuscitation, ice packs placed on the axillae/groin/neck and a cooling blanket placed over the body. In addition, Case One was given a broad-spectrum antibiotic in the possible event of sepsis from an unknown origin and a further bolus of 800 mcg naloxone without effect.

Medication history at the time of the heatstroke included Benzatropine 2 mg tablet taken four times daily, Amisulpride 200 mg tablet taken twice daily, Amlodipine 5 mg + Olmesartan 40 mg tablet taken once daily and one dose of Aclidinium 322 mcg inhaled twice daily.

Case One’s social and economic conditions consisted of experiencing secondary homelessness without family support, and couch surfing at a friend’s property for the last 16 years where he sleeps on a mattress on the floor. Case One has been reported to the Police for welfare follow-up after being away for periods of time (reported in clinical notes—social worker).

Laboratory tests were drawn, showing elevated levels of creatinine of 2.21 umol/L, urea of 9.8 mmol/L and cortisol of 1820 nmol/L. Troponin levels were high at 16,600 ng/L and creatine kinase (CK) was mildly elevated at 155 U/L. The CK level peaked at 1330 U/L on day 2 of admission which could have partly been due to rhabdomyolysis which is commonly seen in heatstroke. Case One had a high limit of the normal level of potassium with 5.2 mmol/L. The blood test also showed evidence of altered liver function with a high aspartate transaminase of 79 U/L, and high alanine transaminase of 77 U/L. Sodium levels were normal for the patient at 136 mmol/L (Table 2). Haematology findings at the time of admission revealed platelets down to 75 10^9/L, a high white blood cell count at 21.9 10^9/L (Table 3) and abnormal coagulation studies. Case One was not diagnosed with disseminated intravascular coagulation. A sepsis screen was completed with both urine and blood cultures negative.

Case One was intubated, ventilated, and admitted to the intensive care unit (ICU) with active cooling (cold intravenous fluids, ice packs on the axillae/groin/neck, and a cooling blanket). Following intubation, a 14 Gauge Foley urinary catheter with a temperature sensor was inserted, and the core temperature reading was 40.5 °C. Between the second day of admission and two weeks from the event, Case One presented a normal trans-oesophageal echocardiogram (TOE), Normal Left Ventricular function, CT angiography and CT Brain with no abnormality detected (NAD). Case One required inotropic infusion for six days, one day of continuous veno-venous hemodialysis (CVVHD) and was intubated for seven days.

Upon discharge from the ICU to the acute rehabilitation unit, Case One was noted awake but drowsy and was able to say his name but unable to recall his living location or any people to contact. After returning to the ward owing to deconditioning, he was transferred to rehabilitation on 17 December and discharged on 23 December. Upon discharge from the hospital, Case One was back to baseline functioning post a rehabilitation stay. Case One was hospitalised for 26 days and the total associated medical costs to the hospital were $66,537 AUD.

### 3.2. Case Two

On the 29 November 2020, a 55-year-old male was found by the police to be drowsy and sitting outside drinking alcohol. An ambulance was called to the scene and paramedics identified the patient to be intoxicated, ataxic, lethargic, sweaty, and wearing four layers of clothing (time of admission to ED 13:16). Case Two’s vital signs at presentation were as follows: T of 38.3 °C measured with an infrared tympanic thermometer; BP of 127/89 mmHg; and HR of 130 bpm (sinus tachycardia). Case Two was drowsy with a GCS score of 13 (Eyes response = 3, Verbal response = 4, Motor response = 6) and was diagnosed with heat exhaustion. Case Two’s main clinical data is presented in Table 1.

Case Two reported being homeless for 7–8 years and was accessing accommodation through the department of housing but reported sleeping rough due to loneliness at the time of the incident. In addition, Case Two has a substance use disorder (Alcohol, Benzodiazepines, Opiates, Amphetamines) with a history of alcohol withdrawal seizures. Past medical history includes liver cirrhosis (Child-Pugh class C) and untreated Chronic Hepatitis C virus. Case Two had splenomegaly, oesophageal varices, and a transjugular intrahepatic shunt in 2016 and had been admitted for previous episodes of hepatic encephalopathy. Case Two also had a history of depression. Case Two’s reported medications at the time of the incident were a Mirtazapine 30 mg tablet taken once daily and a Thiamine 300 mg capsule taken once daily.

The initial management consisted of removing clothes, intravenous fluids, and intravenous administration of 500 mg Thiamine and Diazepam/Oxazepam loading in the context of alcohol withdrawal seizure history. Cold compressors were applied to the groin, armpits and neck.

At the time of admission to the ED, laboratory tests revealed normal levels of creatinine of 69 umol/L and urea of 4.0 mmol/L. The potassium level was also normal at 3.8 mmol/L. There was evidence of altered liver function with a high aspartate transaminase of 67 U/L and alkaline phosphatase of 204 U/L. Sodium levels were high at 149 mmol/L (Table 2). According to the clinical records, liver function tests have shown similar results in the previous two years. No neurology consistent with hepatic encephalopathy was found on examination. The small abnormalities in haematological parameters are shown in Table 3.

The next day, Case Two was reviewed by the drug and alcohol team and subsequently admitted to a rehabilitation facility on the 30 November 2020 under an addiction consultant. Case Two was examined by addiction medicine specialists and no documentation of suspicion of hepatic encephalopathy diagnosis was recorded. Case Two was discharged on the 3 December 2020 after 5 days of hospitalization. The total associated medical costs to the hospital were $3647 AUD.

## 4. Discussion

Two cases of heat illness are reported in this paper, both requiring emergency care in people experiencing homelessness during an early heatwave in November 2020 in Sydney, Australia. The first case presented with classic heatstroke resulting in 26 days of hospitalisation and $66,537 AUD in hospital expenditure. This individual will likely be at increased risk of ongoing health complications that are common in survivors of heatstroke [4] and may be at higher risk of mortality relative to other people experiencing homelessness (which is already higher relative to those not experiencing homelessness [30]). The second case presented with heat exhaustion and spent 5 days hospitalised, costing $3647 AUD. These two cases serve to demonstrate that heatwaves are a significant health threat to people experiencing homelessness and highlight the requirement for further research aimed at protecting this vulnerable population.

It is reported that the most vulnerable people to heat-related illness during heatwaves are elderly individuals living independently with limited or no access to air conditioning. Indeed, there are a greater proportion of elderly individuals with low social support presenting to EDs with heat-related illness relative to young-adult populations [12]. Unfortunately, there is limited research on how heatwaves affect homeless people’s health. Many people experiencing homelessness also live alone and without access to air-conditioning. Moreover, those sleeping rough are continually exposed to outdoor temperatures, usually in an urban environment [24]. This is problematic considering cities can act as a heat sink, absorbing heat during the day with higher temperatures persisting through the evenings, creating an urban heat island effect [26]. This exposure is likely to further increase the risk for this vulnerable population. Prevention aside, timely recognition and treatment are essential for reducing morbidity and mortality from classic heatstroke. However, classic heatstroke signs and symptoms such as confusion, dizziness, headaches, fainting, delirium and lethargy either present suddenly or progressively [4]. The signs and symptoms usually go unnoticed by the sufferer, but rather are recognised by bystanders—as for the two cases presented in this paper. Hence, there is a high risk of heatstroke for people living independently, and an even greater risk for those independent people sleeping rough who are more exposed to outdoor day and night temperatures.

Epidemiological research shows a strong positive association between taking medications such as antipsychotics, antidepressants, anticholinergics and antihypertensives and the risk of morbidity and mortality during extreme heat events [22,23]. It was reported that case one was taking an antipsychotic, anticholinergic and calcium channel blocker, and case two was taking an antidepressant. These medications may have contributed to the heat-related illness through a reduction in autonomic heat loss responses, i.e., a reduction in skin blood flow (limiting dry heat loss) and a reduction in sweating (limiting evaporative heat loss) [31]. While there is some experimental evidence linking certain medications with blunted autonomic heat loss responses, the effect of medications on thermoregulation needs further empirical evidence, garnered from research using realistic doses in realistic environmental conditions [32]. Furthermore, it is important to consider that while physiological impairments may occur, behavioural thermoregulation (i.e., taking actions to keep cool, such as to seek shade/cooler environment, remove clothing, drink water, water douse, etc.) is the most powerful tool humans have to prevent dangerous rises in core temperature [33]. Indeed, it was reported that case two presented to ED in four layers of clothing, suggesting that even if there was a reduction in the ability to autonomically thermoregulate, this was unlikely to be the primary contributing factor to the development of heat-related illness. Rather, it was a reduced capacity to behaviourally thermoregulate, resulting in the case of wearing the four layers of clothing which created a barrier to prevent dry and evaporative heat loss and the resulting dangerous rise in core temperature.

Prevention is by far the most effective option to reduce heat-related morbidity and mortality. At a fundamental level, when people do not have access to air conditioning or other more cost-effective evidence-based cooling methods during heatwaves, they may gain more heat than they can lose. Hence, their core temperature continues to increase, and this is known as uncompensable heat stress. It is highly likely case one and two were experiencing uncompensable heat stress that could have been prevented. The most effective prevention strategies require early heatwave prediction and implementation of cooling interventions for at-risk individuals to prevent or minimise this rise in core temperature. When climate-controlled spaces are unavailable, other cooling strategies are required. However, many heat-policies developed by leading government organisations to protect against heat-related illness lack evidence for their guidance on these cooling strategies [34]. As an example, the heat index (HI) expresses the combined effects of ambient temperature and humidity. Using an electric fan over an HI of 37.2 °C was reported to increase heat stress by the U.S. Environmental Protection Agency [35]. Yet, empirical evidence shows that in 40 °C and 50% relative humidity, which is an HI of 56 °C, fans reduced heat and cardiovascular strain in young healthy men and are reported to be useful when relative humidity is >30% [36].

Cost-effective evidence-based cooling methods such as sitting in front of an electric fan [36] and water-dousing [37] can mitigate rises in core temperature and cardiovascular strain in hot and humid conditions. Other methods of mitigating thermal and/or cardiovascular strain in hot and dry or moderately humid conditions include cold-water foot baths [37] and wearing water-soaked t-shirts [38]. Policies and guidelines around preventing heat-related illness need to be updated to reflect current research findings—specifically guidance on what intervention is most effective at various ambient temperature and relative humidity ranges [34,36,37]. Furthermore, heat-policies should provide practical strategies to increase social support to vulnerable individuals who may require assistance in administering these cooling interventions and early recognition of signs of heat illness. Ideally, if case two received extra social support it is likely they would have been instructed to remove the extra layers of clothing which was likely creating the barrier to heat loss.

Due to the likely higher risk of heat-related illness during a heatwave for people experiencing homelessness, appropriate policies need to be actioned early in response to forecasted heatwaves. The NSW State heatwave subplan of the NSW State Emergency Management Plan [39] is initiated at times of extreme heat. This is a broader action plan that outlines required responses including forecasting, reporting, and planning from a range of departments [39]. The NSW Health Inner City Emergency Response Protocol [40] for people sleeping rough outlines the various actions required by staff to prevent and minimise morbidity and mortality in this at-risk population during severe weather events. During heatwaves, water, sunscreen, hats and increased social support are provided to those sleeping rough. Access to a “heat hub” is also suggested however is yet to be established at the time this paper was published. Unfortunately, the policies and responses detailed above for this Sydney hospital were not sufficient to prevent the heat illness experienced by the two cases presented.

Further actions can be taken to provide greater protection from extreme heat events for people experiencing homelessness. These actions may include delivering extra water to self-douse (not just consume) and cooling packs to apply to the neck, groin, and axillae when staff are on extra outreach shifts. Furthermore, staff on outreach could target areas with a higher density of vulnerable rough sleepers and be trained to recognise early signs of heat illness and how to respond. Establishing a cooling hub as outlined in the NSW Health Inner City Emergency Response Protocol for people sleeping rough could be preventative so long as evidence-based cooling interventions are provided based on the prevailing temperature and humidity. On most hot weather days in Sydney, Australia [34], these interventions could include providing shade, electric fans, misting fans, cold water foot baths, ice towels (ice wrapped in wet towels and placed around the neck) and extra water for drinking and dousing. If more than one hub can be established in major cities more people experiencing homelessness will have access and protection. Secondary prevention should also be practiced whereby those who have experienced heat-related illness, such as the cases presented, are educated on the factors that contributed to the incident and actions to take in the future (cooling interventions outlined above) to prevent it—this was not reported to have occurred in these cases.

Establishing cooling hubs for people experiencing homelessness in a major city such as Sydney would require comprehensive planning across multiple stakeholders, not just for implementation but also utilisation. Cooling centres have been used widely with significant learnings gained—see Widerynski et al., 2017 [41]. The “build it and they will come” approach for cooling centres is flawed, with minimal awareness [41] among other barriers resulting in underutilisation. Awareness and utilisation can be increased with targeted communication strategies such as announcements across media outlets to inform the public of forecasted heatwaves and specific cooling centre locations [41]. To notify those most vulnerable, such as those sleeping rough, community outreach [41] might be the most effective option. Pamphlets and brochures could be printed and distributed by homelessness outreach teams and posters displayed in areas known to have a high population of rough sleepers. Staff on outreach could also educate those sleeping rough on the dangers of excessive heat exposure, common signs of heat illness and how to stay cool if they are unable to access the cooling hub, e.g., seek shade and water douse. Other foreseeable barriers to accessing a cooling hub for people experiencing homelessness may include a lack of transportation to the hub, the fear of leaving personal items or pets unattended [41] and the lack of thermal perception which is the key trigger for behavioural thermoregulation (seek out a cooling hub) [42]. These barriers could be overcome by setting up cooling hubs in pre-determined locations with easy access for rough sleepers along with courtesy transport, providing extra space for belongings and pets and extra outreach services to guide people to each cooling hub. Planning, designing, and implementing cooling hubs that utilise existing public services, personnel and inexpensive evidence-based cooling strategies is a low-cost strategy that can potentially save lives and reduce the health-related economic burden experienced during heatwaves. Despite limited evidence on the direct health and economic impacts of cooling centres they are still extensively used across the United States, Canada, and Europe [41]. Of the facilities that provided space as a cooling centre during heatwaves, 62% reported no additional costs, while 23% reported additional staff hours and 17% reported bottled water as additional costs [41]. Temporary accommodation for people sleeping rough could be another option for protection from the heat, with most accommodation providing air-conditioning. One night in temporary accommodation in New South Wales, Australia, costs on average ~$221 AUD [43], so ~$663 AUD over a 3-to-4-day heatwave (3 nights). Hypothetically, if the two cases presented in this paper attended a cooling hub or cooling centre or received temporary accommodation over the November 2020 heatwave it is likely they could have been spared from the heat illness they suffered, and the hospital could have saved $70,184 AUD (not including the temporary accommodation costs outlined above or any estimated costs of establishing cooling hubs—see limitations below).

Our findings are limited to two cases and as such are not to be generalized. Indeed, this is the reason a more representative sample and more research on heat-related illness in this vulnerable population is needed. We hope that this case series report raises awareness, allowing for the detection and reporting of similar cases or moreover, prevention of similar cases during heatwaves. Another limitation to this report is the lack of specific costing to establish a cooling hub and hence the inability to present the real potential and cost-effectiveness of the intervention. Further research should aim to determine the environmental exposure and incidence of heat-related illness for people experiencing homelessness, as well as the effectiveness of cooling hubs at reducing the incidence.

## 5. Conclusions

There is limited research on how extreme weather affects homeless people’s health, and more funding is required to determine risks and strategies for protection. It is evident from this case study series that severe weather can not only be detrimental to homeless people’s health but can also cause a significant economic toll, evident by the $70,184 AUD expenditure on the care for these two cases. Increased monitoring and a prompt response from public health authorities to ensure effective and timely implementation of evidence-based, low-cost cooling strategies through community outreach and designated cooling hubs could be part of protecting people experiencing homelessness during heatwaves. Furthermore, when responding to heatwaves, it may be necessary to develop plans to reach people experiencing homelessness by considering factors that identify those at the highest risk of a heat-related illness at the individual and community levels. However, these actions will not come without major challenges and will require collaboration across many stakeholders, as well as a thorough cost–benefit analysis.

## Figures and Tables

**Table 1 ijerph-19-16565-t001:** Main clinical data at time of admission.

Clinical Data	Case 1	Case 2
Age (years)	66	55
Sex	Male	Male
Co-morbidities	Schizophrenia, hypertension, NK leukaemia, emphysema, heavy smoker	Liver cirrhosis, chronic hepatitis C virus, depression, alcohol use disorder
Signs and symptoms	Unconsciousness, febrile, hypoventilation	Drowsy, tachycardic, febrile, intoxicated
Core temperature (°C)	41.3 (Tympanic) 40.5 (Urethra)	38.3 (Tympanic)
Respiratory Rate (bpm)	8	16
SpO_2_ (%)	99	89
Heart Rate (bpm)	138	130
SBP (mmHg)	80	127
DBP (mmHg)	-	89
BSL (mmol/L)	10.3	6.7
GCS	3	13
Diagnosis	Heatstroke	Heat Exhaustion

GCS—Glasgow Coma Scale; BSL—blood sugar level; SpO_2_—oxygen saturation (pulse oximeter); SBP—systolic blood pressure; DBP—diastolic blood pressure; NK—natural killer.

**Table 2 ijerph-19-16565-t002:** Biochemistry findings at the time of admission.

Biochemistry	Case 1 (Normal Range)	Case 2 (Normal Range)
Total protein (g/L)	75 (60–80)	61 (60–80)
Albumin (g/L)	46 (33–48)	39 (33–48)
Aspartate transaminase (U/L)	79 *^H^ (0–35)	67 *^H^ (0–35)
Alanine transaminase (U/L)	77 *^H^ (0–40)	29 (0–40)
Alkaline phosphatase (U/L)	64 (30–110)	204 *^H^ (30–110)
Creatinine (µmol/L)	221 *^H^ (60–110)	69 (60–110)
Total bilirubin (µmol/L)	11 (0–20)	31 *^H^ (0–20)
Sodium (mmol/L)	136 (135–145)	149 *^H^ (135–145)
Potassium (mmol/L)	5.2 (3.5–5.2)	3.8 (3.5–5.2)
Magnesium (mmol/L)	0.75 (0.70–1.10)	0.83 (0.70–1.10)
Calcium (mmol/L)	2.62 *^H^ (2.10–2.60)	2.10 (2.10–2.60)
Chloride (mmol/L)	102 (95–110)	110 (95–110)
Bicarbonate (mmol/L)	15 *^L^ (22–32)	23 (22–32)
Urea (mmol/L)	9.8 *^H^ (4.0–9.0)	4.0 (4.0–9.0)
Cortisol (nmol/L)	1820 *^H^ (150–520)	Not available
Troponin (ng/L)	16600 *^H^ (0–20)	Not available
Creatine Kinase (U/L)	155 ^#^ (20–250)	224 (20–250)
Blood alcohol concentration (g%)	<0.03 (0.00)	0.12 (0.00)

*^H^—Indicates high level; *^L^—Indicates low level. ^#^ CK peaked at 1330 U/L on day 2 of admission.

**Table 3 ijerph-19-16565-t003:** Haematology findings at the time of admission.

Complete Blood Cell Counts	Case 1 (Normal Range)	Case 2 (Normal Range)
Red blood cells (10^12/L)	5.1 (4.5–6.5)	4.5 (4.5–6.5)
Haemoglobin (g/L)	164 (130–180)	126 *^L^ (130–180)
Platelet (10^9/L)	75 *^L^ (150–400)	121 *^L^ (150–400)
White blood cells (10^9/L)	21.9 *^H^ (4.0–11.0)	5.9 (4.0–11.0)
Neutrophils (10^9/L)	16.2 *^H^ (2.0–7.5)	4.1 (2.0–7.5)
Lymphocytes (10^9/L)	2.6 (1.5–4.0)	0.9 *^L^ (1.5–4.0)
Monocytes (10^9/L)	2.8 *^H^ (0.2–1.0)	0.7 (0.2–1.0)
Eosinophils (10^9/L)	0.0 (0.0–0.4)	0.1 (0.0–0.4)
Basophils (10^9/L)	0.0 (0.0–0.1)	0.1 (0.0–0.1)
**Haemostasis**		
Prothrombin time PT (Seconds)	15 (11–15)	Not available
International normalised ratio	1.1	1.8
Fibrinogen (g/L)	1.8 *^L^ (2.0–4.0)	Not available
D-dimer (mg/L)	2.54 *^H^ (<0.5)	Not available
**Blood Gases**	**Arterial Blood**	**Venous Blood**
pH	7.16 *^L^ (7.35–7.45)	7.36 (7.32–7.42)
PaCO_2_ (mmHg)	59 *^H^ (32–45)	43 (38–52)
PaO_2_ (mmHg)	213 *^H^ (75–105)	92 *^H^ (29–48)
HCO_3_^−^ (mmol/L)	20 *^L^ (24–31)	24 (24–31)
Base excess	–10 *^L^ (–3–3)	–1 (–3–3)
Lactate (mmol/L)	2.5 *^H^ (0.0–2.2)	2.9 *^H^ (0.0–2.2)

*^H^—Indicates high level; *^L^—Indicates low level; FiO_2_—estimation of the fraction of inspired oxygen; pH—potential hydrogen; PaCO_2_—partial pressure of carbon dioxide; PaO_2_—partial pressure of oxygen; HCO_3_^−^—bicarbonate.

## Data Availability

Not applicable.

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
