# Peer review of "Heat Illness Requiring Emergency Care for People Experiencing Homelessness: A Case Study Series"

_ijerph, 2022, doi:10.3390/ijerph192416565_

Round 1
Reviewer 1 Report
Thank you for the opportunity to review this paper. Overall, the paper is interesting, but I do believe there are both ethical concerns that need to be addressed. Suggested amendments are as follows:
1. Please amend the term ‘alcoholism’ to alcohol use disorder. Alcoholism is considered a stigmatising term, whereas alcohol use disorder is the standard DSM-V description.
2. The first line in the introduction (“Global warming...”) needs a direct reference for support.
3. Page 2, line 94 – was the waiver of consent awarded by an ethics committee? This section should specify whether the project underwent ethical review, particularly given the nature of the detail provided. The detail provided means that your case studies could be easily identified to both healthcare and support workers familiar with the case, hence why I believe an ethical review would be necessary.
4. In most journals, it is customary for case studies to be read by the patient prior to publication – some journals will not allow publication without agreement. Did this occur? Again, I am concerned that many of these details allow ready identification for those who have worked with these individuals, which may indicate the need for informed consent for these details to be published.
5. I would suggest removing terms used in clinical notes (i.e. “nil effect”). Perhaps state “no effect was noted from...” or similar.
6. I would suggest avoiding terms like ‘drug dependency’ and using terminology aligned with the DSM-V (use disorder) or simply stating use of (drug).
7. There are parts of the discussion that could be tied more closely to the cases presented – for instance, the section on educating individuals to protect themselves from extreme heat events. Did this occur in these cases? It would be interesting to see the challenges of providing heat education to the cases presented here, as you have done with the next paragraph on heat refuges.
8. Please outline the limitation of a case study of two individuals in your discussion.
9. I think your conclusion should be a paragraph rather than a numbered list. There are some very strong key points in this paper (challenges and cost especially) that would be communicated more effectively in a paragraph.
Author Response
7th December 2022
Dear Reviewer
I appreciate your consideration of this paper titled ‘Heat illness requiring emergency care for people experiencing homelessness: a case study series'. We have reflected on the reviewer's comments and are grateful for their thorough review of our paper. Please see the attachment that identifies our response to each reviewer's comments.
I appreciate your consideration of this submission,
Yours Sincerely,
Alejandro Vasquez Hernandez | Research Officer
Homeless Health Service
St Vincent’s Hospital Sydney | 390 Victoria Street DARLINGHURST NSW 2010
T 02 8382 1484
M 0450 707 129
E alejandro.vasquez@svha.org.au

Reviewer 2 Report
This is an interesting case series that describes heatwave-related illness among homeless people. However, the novelty and importance of the report is still not clear. So I think the following revision might be needed.
>The authors may need to show the (estimated) number of patients who are affected by heat-related illness in Australia and total cost of the treatment so that the readers can see the importance of the intervention.
>The estimated cost of establishing cooling hub is also needed to show the cost-effectiveness of the intervention.
>Please show the temperature and humidity at the point when the patients were transferred. Usually measure of heat stress uses WBGT, so if the authors can calculate the WBGT, it would be better.
>In case one, the increased creatinine and CK seem to be partly due to rhabdomyolysis rather than myocardial injury, which is common in heat-shock. Please discuss this point.
>In case two, even thou the hepatic function was not changed, differential diagnosis may still include hepatic encephalopathy, which is often caused by accumulation of ammonium or lack of vitamin B1. The patient showed increased lactate but the Ph was within normal, which suggest potential alkalosis might exist. Please discuss this point, and also please present the titre of NH3 , vitamin B1, if possible. In addition, please describe whether vitamin and BCAA were supplemented at that time.
Author Response

(The authors gave the same response as above.)

Round 2
Reviewer 1 Report
Thank you for attending to the revisions, your response is very comprehensive and in my opinion has improved the paper. I suggest accepting the paper (subject to the satisfaction of the second reviewer).
Reviewer 2 Report
I think the authors responded to my comments well. Thank you.